# StrucBooth: Structural Gradient Supervised Tuning for Enhanced Portrait Animation

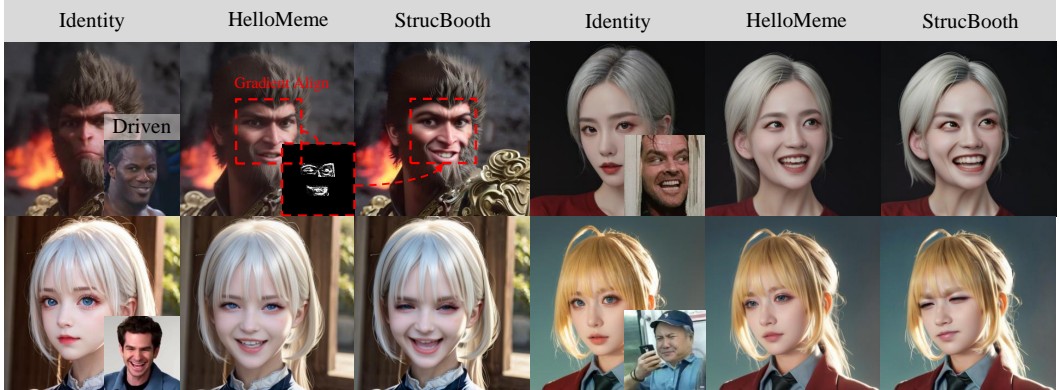

Figure 1: **Visual Results**. StrucBooth extracts and incorporates expression structure details from the pixel space into the model, enabling the optimization of facial expressions.

## Abstract

Portrait animation aims to synthesize images or videos that transfer expressions or poses from a reference while preserving identity. Existing methods often rely on high-level expression encoders, which capture only coarse semantics and miss fine-grained structural details in critical regions such as the eyes, eyebrows, and mouth, leading to noticeable discrepancies and suboptimal expression fidelity. To address this, we propose `StrucBooth`, a framework that binds pixel-level expression structures into the model through case-specific optimization while preserving the generator's inherent capabilities. `StrucBooth` combines (i) PGT-based self-tuning, which uses a preliminary prediction as Pseudo Ground Truth (PGT) for lightweight refinement, and (ii) pixel-level structural supervision, which extracts gradient variations (Facial Structural Gradients) from expression-related patches and aligns them to inject fine-grained structural information. Extensive evaluations under both cross-driven and self-driven settings demonstrate that `StrucBooth` consistently improves expression accuracy over strong baselines, highlighting that integrating pixel-space structural signals is an effective direction for faithful and visually consistent portrait animation.

## 1 Introduction

With the rapid advancement of AIGC technologies, image and video generation has achieved remarkable progress. State-of-the-art methods can now synthesize highly realistic content (BlackForest, 2024; Esser et al., 2024; Kong et al., 2024; Yang et al., 2024b; Wan et al., 2025), enabling a wide range of applications such as personalized content creation and creative media production. In this context, the intrinsic human desire to experience novel visual content has motivated the development of the Portrait Animation task (Xu et al., 2025; Xie et al., 2024; Zhang et al., 2024a; Zhao et al., 2025; Guo et al., 2024b). The goal of this task is to generate images or videos in which a target identity faithfully reproduces the expressions, poses, and motions of a reference source, which may be provided as either a single image or a short video clip.

Although recent advances have enabled the synthesis of visually convincing portraits, existing works (Guo et al., 2024b; Wang et al., 2024a; Zhang et al., 2023b; Drobyshev et al., 2023) often fall short in faithfully capturing fine-grained expression structures. Rich structural details and subtle variations across different expressions are difficult for general-purpose expression synthesis models to preserve, leading to noticeable discrepancies in fine-scale features. In addition, most methods rely on 2D or 3D expression representations of feature level (Doukas et al., 2021; Yin et al., 2022; Huang et al., 2023; Yu et al., 2023; Khakhulin et al., 2022; Mi et al., 2024; Tao et al., 2024), which often compress or omit structural details in pixel space, limiting the model's ability to perceive these structures. Therefore, we aim to *enhance the existing generative capabilities* by extracting and incorporating *pixel-level expression structural details* from specific expression cases, enabling fine-grained expression optimization.

We thus propose `StrucBooth`, a framework designed to **incorporate pixel-level expression structures details into the model** through case-specific optimization while preserving the inherent generative capabilities of the generator. `StrucBooth` consists of the following two components: **(i) PGT-based self-tuning:** As shown in Fig. 2(a), for each input case, we first generate a preliminary prediction using the generator to be optimized, which serves as the **Pseudo Ground Truth (PGT)**. The PGT provides self-supervision for the following finetune stage, preserving the generator's basic generative ability. **(ii) Facial Structural Gradients supervision:** We posit that structural information derived directly from the pixel space offers rich cues for expression imitation. To this end, we extract Facial Structural Gradients, capturing the gradient variations of pixels. These gradients convey strong structural signals while minimizing the influence of irrelevant factors such as color and appearance. During optimization, we align the gradients of image patches corresponding to expression-relevant regions, effectively incorporating fine-grained structural information into the model.

Our experiments show that the proposed framework substantially improves expression similarity while preserving identity. For instance, on Hellomeme (Zhang et al., 2024a), EXP_SIM (Zhao et al., 2025) increases from 0.2572 to 0.3185 with a corresponding drop in AED (Siarohin et al., 2019), and on HunyuanPortrait (Xu et al., 2025), it improves from 0.2886 to 0.3265. In addition, self-bench results reveal gains in SSIM (Wang et al., 2004) and PSNR, confirming enhanced structural consistency without loss of perceptual quality. These results demonstrate that a few hundred optimization steps, combined with structural supervision, are sufficient to refine expression fidelity while maintaining output reliability.

In summary, our contributions are as follows:

- We propose `StrucBooth`, a framework that *incorporates pixel-level expression structure details* into the model through a few-step self-tuning process using PGT, optimizing expression structures while preserving the model's inherent generative capabilities.
- We introduce **Facial Structural Gradients** supervision, by aligning structural details in the pixel space, we enhance the fine-grained consistency of expressions in the generated results.
- We conducted extensive experiments in multiple baselines and evaluation protocols. Our method consistently improves expression fidelity and consistency.

## 2 RELATED WORK

### 2.1 DIFFUSION MODELS FOR VISUAL SYNTHESIS

Diffusion models have become a dominant generative paradigm for visual content creation, surpassing adversarial and autoregressive approaches in both fidelity and diversity (Ho et al., 2020; Song et al., 2020; Ho, 2022). By progressively denoising latent variables, they offer stable training and controllable sampling, which has driven major advances in image synthesis. Large-scale text-to-image systems (Saharia et al., 2022; Rombach et al., 2022) further combine diffusion with transformer-based language encoders, achieving strong semantic grounding and photorealistic outputs. To improve structural controllability, methods such as ControlNet (Zhang et al., 2023a; Li et al., 2024b) and IP-Adapter (Ye et al., 2023) introduce conditioning mechanisms based on poses, sketches, and reference images, which extends to more applications (Zhou et al., 2024a;b; Wang et al., 2024c; Liang et al., 2024; Zhou et al., 2024c; 2025; Li et al., 2025; 2024c). Beyond static images, diffusion models have been extended to videos by incorporating temporal modeling into

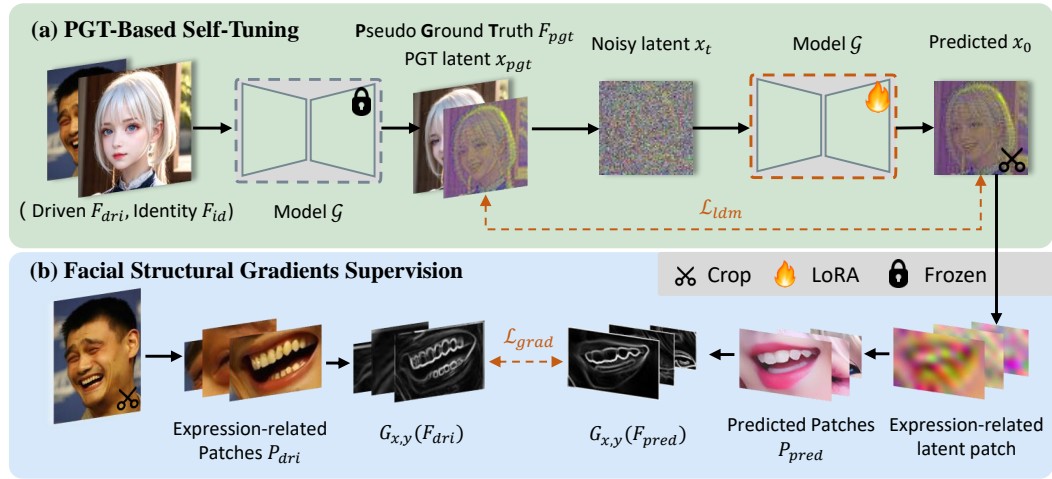

Figure 2: **Overview.** `StrucBooth` first generates pseudo-ground truth (PGT) for self-tuning. During fine-tuning, we align and optimize expressions by aligning the Facial Structural Gradients of expression-related patches between the driving image and the intermediate prediction.

the U-Net backbone (Wan et al., 2025; Kong et al., 2024; Yang et al., 2024b). Factorized spatio-temporal attention (Chen et al., 2023; 2024a) and latent video diffusion (He et al., 2022; Ho et al., 2022b) enable consistent motion generation while maintaining visual quality. Text-to-video systems such as Make-A-Video and Imagen Video (Singer et al., 2022; Ho et al., 2022a) demonstrate strong capabilities in generating short clips directly from textual prompts.

## 2.2 PORTRAIT ANIMATION

Portrait Animation aims to generate realistic motions and expressions from a single static image, typically driven by external signals such as reference videos, landmarks, or audio cues. Traditional non-diffusion approaches primarily rely on explicit motion priors, with 3D Morphable Models (3DMM) (Doukas et al., 2021; Yin et al., 2022; Huang et al., 2023; Yu et al., 2023; Khakhulin et al., 2022; Mi et al., 2024; Tao et al., 2024) being the most common representation. While these methods can reproduce local expressions and mouth movements, their dependence on geometric warping often leads to artifacts under large pose variations, identity mismatch, or occlusions.

The advent of diffusion models (Ho et al., 2020) has opened new directions for portrait animation. Several works (Wei et al., 2024; Yang et al., 2024a; Xie et al., 2024; Guo et al., 2024a) adapt text-to-image diffusion backbones such as Stable Diffusion (Rombach et al., 2022), achieving improved realism and robustness compared to earlier pipelines. To enhance controllability, landmark- or keypoint-guided conditioning (Wang et al., 2024b; Zheng et al., 2024; Chen et al., 2024b) has been introduced, enabling finer expression transfer. However, geometric discrepancies across identities often cause expression misalignment and identity drift. Furthermore, most methods treat generation as a frame-wise process without explicit temporal modeling, which results in flickering and limited temporal smoothness.

Recent advances move toward end-to-end video diffusion frameworks (Zhang et al., 2024b; Peng et al., 2024; Jin et al., 2024; Li et al., 2024a), explicitly integrating temporal coherence into the generative process. While these approaches improve consistency, they typically operate in latent spaces that compress facial motion, inevitably discarding fine-grained structural cues. Such information loss hinders long-horizon consistency and reduces realism for complex expressions, underscoring the need for methods that preserve detailed dynamics in portrait animation.

## 3 METHOD

### 3.1 OVERVIEW

**Task Definition.** The *Portrait Animation* task (Xu et al., 2025) takes as input an identity image $F_{id}$, which specifies the static appearance of the target person, and a driving image or video $F_{dri}$ (hereafter referred to as a frame), which encodes the desired expressions, head motions, and poses. The objective is to generate a new frame $F_{gen}$ that faithfully preserves identity-specific details (e.g., facial geometry, hairstyle, background) from $F_{id}$, while accurately transferring the dynamic expression attributes from $F_{dri}$.

**Method Overview.** We conceptualize expression as an attribute on par with identity, and build on the idea of incorporating it into the model through few-steps optimization with pixel-level expression structural supervision. As illustrated in Fig. 2, we propose the `StrucBooth` framework. Concretely, we introduce a case-specific PGT-tuning scheme, where the generator is finetuned for a few steps using generated **Pseudo Ground Truth (PGT)**, enabling it to preserve the original generative capabilities for identity and background within each case. Meanwhile, during the finetuning process, we extract **Facial Structural Gradients** (see details in Sec.3.3), a pixel-level expression structure details, directly from the driving frame $F_{dri}$. By focusing on expression-relevant regions, we align the intermediate predictions with the corresponding areas of the driving frame, guiding fine-grained, case-specific optimization of expression structures.

### 3.2 PGT-BASED SELF-TUNING

**PGT Generation.** As previously mentioned, we aim to capture the pixel-level expression structure details for each case via LoRA. An important issue is that, for any given case, the finetuning process lacks true supervision for the post-imitation expression. So we propose the PGT self-tuning, which preserves the model's core generative capabilities.

We first synthesize a PGT image to serve as the pseudo-ground truth for the current case. Specifically, as shown in the left half of Fig. 2 (a), given a generator $\mathcal{G}$ and an arbitrary input case ($F_{dri}, F_{id}$), we first use $\mathcal{G}$ to produce an initial prediction $F_{pgt}$:

$$F_{pgt} = \mathcal{G}(F_{dri}, F_{id}) \tag{1}$$

**Self-tuning.** Next, we establish a basic finetuning paradigm in the right half of Fig. 2 (a): We first encode the $F_{pgt}$ frame in the VAE latent $x_{pgt}$, then add Gaussian noise $\epsilon$ at a given timestep $t$, and feed the noisy latents $x_t$ into the model to predict the added noise or the corresponding flow:

$$\mathcal{L}_{ldm} = \mathbb{E}x_t, \epsilon \sim \mathcal{N}(0,1)\Big[\big|\epsilon - \epsilon_\theta(x_t + \epsilon)\big|_2^2\Big], \tag{2}$$

Here, $x_t$ denotes the noisy PGT latent, $\epsilon_\theta$ represents the model's prediction of the noise. If training on flow instead, $\epsilon$ can be replaced with the ground-truth flow, while the formula remains in the form of an MSE loss. This optimization step preserves the model's ability to generate the pose, identity, background, and expression for the current case, preventing excessive distribution shift.

### 3.3 FACIAL STRUCTURAL GRADIENTS SUPERVISION

**Facial Structural Gradients.** During the optimization process, we further incorporate pixel-level structural supervision to bind the expression structure of the driving frame $F_{dri}$ to the model. While the driving frames $F_{dri}$ contain rich expression-related structural information, they also differ from the provided identity images $F_{id}$ in terms of appearance like identity, sharpness, and color. Therefore, we focus on extracting a structural representation that emphasizes expression-relevant details while being robust to variations in appearance. We emphasize that *expressions are primarily manifested through distortions and deformations of facial structures*, and therefore adopt image gradients as attribute-agnostic structural guidance. As shown in Fig. 2(b), gradients capture pixel-wise variation trends and convey clear, recognizable expression information even without color or other appearance cues. They provide strong structural signals and effectively emphasize subtle details that are critical for accurate expression representation. Therefore, we extract gradients using the Sobel

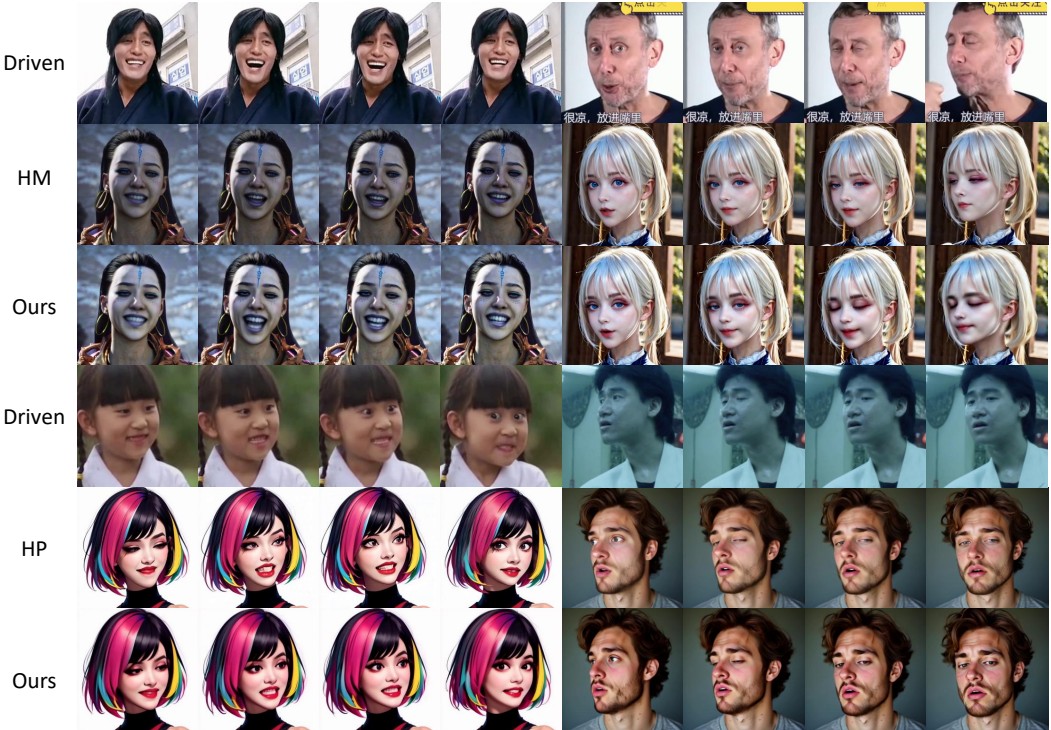

Figure 3: **Results in Cross-Video bench.** Top 3 lines indicating results with HelloMeme(HM) as baseline. Bottom 3 lines indicating results with HunyuanPotrait(HP) as baseline.

operator. Concretely, we first convert the input frame $F$ to grayscale using a function $\mathcal{F}_{gray}(\cdot)$ to remove color information. We then apply Sobel convolution kernels $K_x$ and $K_y$ to compute horizontal and vertical gradients at each pixel, forming local gradient vectors that capture directional changes and fine structural patterns of facial features:

$$\mathbf{G}_{x,y}(F) = \begin{bmatrix} G_x(F) \\ G_y(F) \end{bmatrix} = \begin{bmatrix} (K_x * \mathcal{F}_{gray}(F)) \\ (K_y * \mathcal{F}_{gray}(F)) \end{bmatrix}, \tag{3}$$

where the Sobel kernels are defined as:

$$K_x = \begin{bmatrix} 1 & 0 & -1 \\ 2 & 0 & -2 \\ 1 & 0 & -1 \end{bmatrix}, \quad K_y = \begin{bmatrix} 1 & 2 & 1 \\ 0 & 0 & 0 \\ -1 & -2 & -1 \end{bmatrix}. \tag{4}$$

**Patch-Wise Structure Alignment.** In a single step of the finetuning process, given the model's predicted noise or flow $\epsilon$, we first compute the predicted $x_0$ and decode it through the VAE to obtain the predicted pixel frame $F_{pred}$. We then extract the Facial Structural Gradients as described above from both the driving frame $F_{dri}$ and the predicted frame $F_{pred}$, and try to optimize the expression by aligning the gradient details between the two frames. However, directly aligning gradients between the two images ($F_{dri}, F_{pred}$) is impractical, since the driving frame and the predicted frame correspond to different identities and thus may exhibit significant differences in the size and spatial position of expression regions such as the eyes.

Given the strongly localized nature of expressions, only specific regions contain structure that is highly relevant to the expression. Therefore, we extract *Expression-relevant patches* from both the driving and predicted frames, resize them to the same size, and perform gradient supervision within these regions. For the predicted frame $F_{pred}$, as shown on the right side of Fig. 2 (b), based on the rough estimates of facial expression regions provided by the PGT, we can detect and crop the corresponding expression-related latents patches from the predicted latents $x_0$ and decode them into patch images $P_{pred}$:

$$P_{pred} = \text{Decode}(x_0 \cdot \mathcal{R}(\text{Detect}(F_{pgt}))). \tag{5}$$

| Driven | Reference | Hellomeme | Ours | Reference | Hellomeme | Ours |
|---|---|---|---|---|---|---|

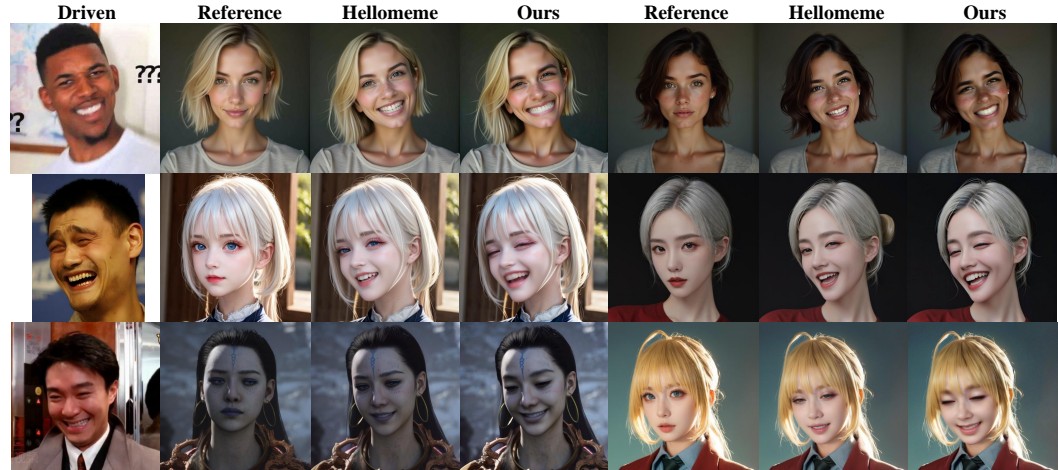

Figure 4: **Results on the image benchmark.** We show the results of Hellomeme on Cross-Image Benchmark.

Here, $\mathrm{Decode}$ denotes decoding the latent patches into images using the VAE, $\mathcal{R}$ represents resizing the masks detected from the PGT to match the latent scale, and $\mathrm{Detect}$ indicates identifying face-related regions using a face detector. Thanks to the spatial perceptual equivalence of the VAE (Rombach et al., 2022), the pixel images decoded from these patches retain high visual quality and preserve the basic structural information, making them suitable for gradient computation and optimization. At the same time, this patch-wise decoding reduces computational overhead by avoiding decoding regions unrelated to the expression, improving efficiency and reducing memory consumption. For the driving frame $F_{dri}$, we detect the locations of expression-relevant regions such as the eyes and mouth using the same face detector, crop these regions, and, guided by the PGT-predicted expression areas, resize the corresponding patches into Expression-related Patches $P_{dri}$ with the same size as $P_{pred}$:

$$P_{dri} = \mathcal{R}(F_{dri} \cdot \mathrm{Detect}(F_{dri})). \tag{6}$$

Based on the spatially aligned patches, we extract the corresponding gradients. To avoid the influence of gradient magnitudes, we align the gradient directions between patches using a cosine loss, achieving pixel-level expression structure alignment:

$$\mathcal{L}_{grad} = \frac{1}{N} \sum_{i=1}^{N} \left( 1 - \frac{\mathbf{G}_{x,y}(P_{pred}) \cdot \mathbf{G}_{x,y}(P_{dri})}{\|\mathbf{G}_{x,y}(P_{pred})\|_2 \, \|\mathbf{G}_{x,y}(P_{dri})\|_2} \right), \tag{7}$$

**Training Loss.** To maintain portrait consistency during training, we additionally introduce a sparse identity (id) loss. In particular, when optimizing videos, identity and related information exhibit strong temporal redundancy. Therefore, we randomly sample one frame from the current video segment, decode it, and compute the identity loss. Consequently, the overall training process involves the following losses:

$$\mathcal{L}_{total} = \lambda_{ldm} \cdot \mathcal{L}_{ldm} + \lambda_{grad} \cdot \mathcal{L}_{grad} + \lambda_{id} \cdot \mathcal{L}_{id}, \tag{8}$$

where $\lambda_{ldm}, \lambda_{grad}, \lambda_{id}$ are the balancing coefficients for each corresponding loss term.

## 4 EXPERIMENTS

### 4.1 SETTINGS

**Benchmarks.** We select HelloMeme (Zhang et al., 2024a) and HunyuanPortrait (Xu et al., 2025) as our baselines, attempt to optimize different cases on top of these methods on the following three benchmarks: *Cross-Image*, *Cross-Video*, and *Self-Video*. Specifically, in the *Cross-Image* benchmark, we manually collected 10 identity images and 20 driving images with pronounced and exaggerated expressions. resulting in a total of 200 test cases. We apply each baseline model to optimize

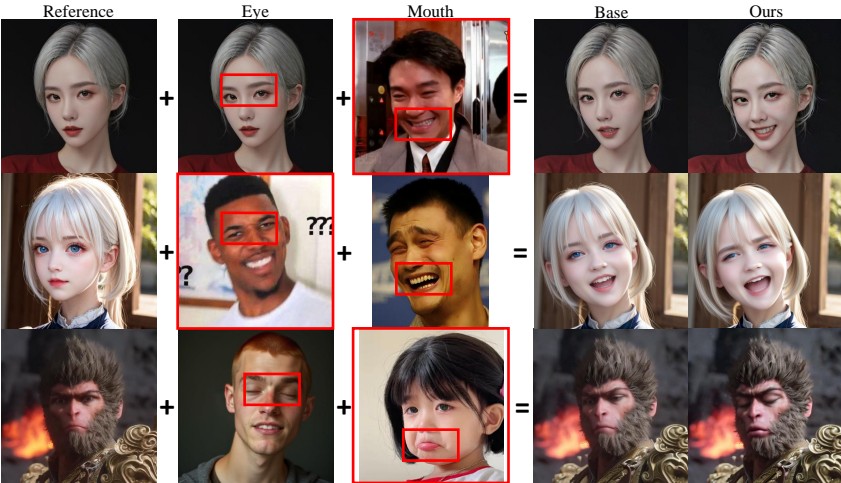

Figure 5: **Application.** StrucBooth support more flexible control and optimization, allowing different expression details to be combined into a single output. **The red box in the figure indicates the source of the output pose.**

expression alignment for these cases. In the *Cross-Video* benchmark, we similarly collect 10 identity images and 20 driving expression videos, yielding another 200 test cases. In the *Self-Video* benchmark, we select 50 short human expression videos (3 seconds each). For each video, we take one frame from the first 10 frames as the identity image, and use the last 2 seconds as the driving expression video.

**Implementation Details.** We adopt the following loss weights in our experiments: We set the loss weights consistently across datasets: the gradient loss weight is $0.1$, the identity-preserving loss weight is $0.1$, and for HelloMeme, the LDM reconstruction loss weight is set to $1.0$. All models are optimized for 800 steps with their default inference configurations on a single 48G NVIDIA A6000 GPU. Please refer to supplementary materials for more details.

## 4.2 METRICS

**Cross-Video/Image Bench.** For the *Cross-Video* and *Cross-Image* benchmarks, we evaluate identity, motion, and expression consistency. Identity Similarity (ID-SIM) is computed using Arc-Face (Deng et al., 2019) embeddings, measuring the cosine similarity between each generated frame and the reference identity. Expression Similarity is assessed via Average Expression Distance (AED) between facial landmarks of generated and driving frames (Siarohin et al., 2019). For the *Cross-Video* benchmark, we also introduce Emotion Similarity (EMO-SIM) (Zhao et al., 2025) to evaluate expression consistency across frames. EMO-SIM employ a pretrained emotion encoder EmoNet (Toisoul et al., 2021) and compute the mean concordance correlation coefficient (CCC) and Pearson correlation coefficient for both valence and arousal. Higher EMO-SIM scores indicate better alignment of subtle expressions and micro-expressions.

**Self-Video Bench.** The *Self-Video* benchmark focuses on pixel-level fidelity. We measure Structural Similarity Index (SSIM) (Wang et al., 2004) and Peak Signal-to-Noise Ratio (PSNR) to quantify alignment of facial expressions. To isolate facial regions, backgrounds are masked prior to metric computation, ensuring that scores reflect improvements in expression synthesis rather than background variations.

## 4.3 QUANTITATIVE AND QUALITATIVE ANALYSIS

**Quantitative Results.** As shown in Table 1, our method consistently improves the EXP_SIM metric (from 0.2572 to 0.3185 on Hellomeme and from 0.2886 to 0.3265 on HunyuanPortrait) while reducing the AED metric across different baselines in the cross-bench setting. This demonstrates

Table 1: **Quantitative evaluation on Cross-Video, Self-Video and Cross-Image benchmarks.** ↑ indicates higher is better, ↓ indicates lower is better. The best results are in bold.

| Method | Cross-Video | | | Self-Video | | Cross-Image | |
|---|---|---|---|---|---|---|---|
| | ID↑ | EXP_SIM↑ | AED↓ | PSNR↑ | SSIM↑ | ID↑ | AED↓ |
| HelloMeme | 0.494 | 0.2572 | 6.37 | 18.89 | 0.78 | **0.721** | 1.16 |
| + StrucBooth | 0.505 | 0.3185 | **5.72** | 19.99 | 0.79 | 0.669 | **1.05** |
| vs. Base | +0.011 | +0.0613 | -0.65 | +1.10 | +0.01 | -0.052 | -0.11 |
| HunyuanPotrait | 0.594 | 0.2886 | 6.66 | 21.54 | 0.83 | - | - |
| + StrucBooth | **0.599** | **0.3265** | 6.55 | **21.83** | **0.84** | | |
| vs. Base | +0.005 | +0.0379 | -0.11 | +0.29 | +0.01 | | |

that, after a limited number of optimization steps, our approach effectively enhances the accuracy of expression simulation. Importantly, the ID metric is largely preserved: fine-tuning toward pseudo-targets maintains identity, with even slight improvements in some cases. In the self-bench setting, our method improves both SSIM and PSNR, indicating enhanced structural consistency in expression imitation without compromising perceptual quality. Overall, these results show that combining a small number of optimization steps with quality-restoration and structural losses allows the model to refine expressions while retaining output fidelity. Furthermore, on HelloMeme, which supports image-level synthesis, our method reduces the AED metric, confirming improved expression similarity. Identity is minimally impacted: the ID metric decreases by only 0.052. These results indicate that, at the image level, our approach effectively captures fine-grained expression details, enhancing the fidelity of expression imitation.

**Quantitative Results.** In Fig. 3, we present the results on the video benchmark. As shown, in the left-hand case, the imitation results exhibit closer similarity in the mouth's range of motion and structural details, making the overall expression more vivid and lifelike. In the right-hand case, our method better captures subtle actions such as half-open or closed eyes, as well as lip-smacking, rendering them more natural and expressive. In Fig. 4, taking the HelloMeme model as an example, we also compare the original generation results with those obtained after approximately 800 optimization steps using our approach. Each row shows results generated from different reference images when driven by the same expression input. We observe that the optimized results display structures around the mouth, eyes, and other expression-related regions that more closely match the driving image, thereby achieving more accurate expression imitation. For example, in the first row, each person's smiling expression better captures the perplexed look from the driving image, with the mouth structure more closely aligned; in the second row, the extent of closed eyes and open mouths is greater, demonstrating improved structural consistency; finally, the grin in the last row is also structurally more faithful. While our method may cause a slight impact on identity preservation, this effect is negligible compared to the substantial improvements in expression quality.

**Application** Patch-based expression optimization enables fine-grained and flexible control, allowing diverse expression details from multiple reference images or videos to be integrated into a single coherent output. To facilitate this, we introduce an intermediate representation, *PasteDrive*, constructed by compositing selected facial regions from different references. Expression optimization is then performed in a patch-wise manner, with each region guided by its own expression gradients. As illustrated in Fig. 5, this region-decoupled design offers two key advantages: (1) it allows selective imitation, enabling the model to replicate only specific components of a reference expression (e.g., the mouth region in the first row); and (2) it supports compositional control, permitting seamless fusion of distinct regions from multiple references (second and third rows). These capabilities underscore the effectiveness of patch-based optimization in achieving precise local control while maintaining flexible expression composition.

## 4.4 ABLATION STUDY

**PGT tuning.** ① with ② in Tab. 2 (a) shows that direct PGT tuning preserves the model's basic generative capability and provides a slight improvement in expression reproduction.

Table 2: **Ablation Results of HelloMeme on Cross-Image.** (a): quantitative ablation table of HelloMeme on Cross-Image. ↑ indicates higher is better. ↓ indicates lower is better. (b): ablation of patch-wise decoding. (c): ablation of optimizing steps.

(b) Patch-Wise decoding

(a) Ablation on HelloMeme

| ID | Method | ID↑ | AED↓ |
|----|--------|-----|------|
| ① | Base | 0.721 | 1.16 |
| ② | ① + PGT | 0.703 | 1.14 |
| ③ | ② + grad | 0.611 | 1.01 |
| ④ | ③ + id | 0.669 | 1.05 |

(c) Optimize Steps

**Facial Structural Gradients.** Compared ② with ③, introducing the gradient loss during PGT tuning effectively reduces the AED metric, enhancing the expression similarity of the model's generated results, but it also leads to a slight decrease in the ID metric.

**Id presevation.** If we optimize using only the gradient loss, as shown in the third row of Tab. 2 (a), the AED can be further reduced (from 1.05 to 1.01), indicating closer expression similarity, but this comes at the cost of increased ID loss. By additionally incorporating an ID-preserving loss, it is possible to recover some ID information without significantly compromising expression imitation. In practice, these two losses represent a trade-off: increasing the weight of the gradient loss favors stronger expression imitation, while increasing the weight of the ID or PGT-tuning loss helps better preserve identity information.

**Patch Wise decoding.** As shown in Tab. 2(b), We show the resource consumption (GPU memory and training speed) and AED metric when computing on single frames. Pentagrams represent patch-wise gradient computation, while circles labeled "Global" denote computing gradients on the full decoded image followed by mask-based filtering. Green values indicate GPU memory usage during execution, and red values indicate computation speed. When only the gradient loss is involved, the patch-wise scheme and the global gradient computation scheme achieve similar AED scores. However, the patch-wise approach consumes fewer resources and improves computation speed by approximately 22%. Moreover, directly optimizing entire videos on a single A6000 GPU is infeasible; by introducing the patch-wise decoding, we can efficiently perform optimization.

**Optimize steps.** As shown in Tab.2 (c), as the number of optimization steps increases, the AED metric gradually decreases, indicating that the expressions become more similar; however, the ID metric also experiences a slight decline. In practice, 400–600 steps are generally sufficient to achieve high expression fidelity without substantially compromising identity preservation.

## 5 CONCLUSION

In this work, we propose `StrucBooth`, a method to optimize expression similarity in portrait animation by extracting structural information directly in pixel space. Our approach can be applied with any expression generator and requires only about 200–800 optimization steps to significantly improve the expression similarity of the generated outputs. By leveraging case-specific PGT-based self-tuning and pixel-level gradient supervision, our method effectively binds fine-grained expression structures into the model while preserving its inherent generative capabilities.Extensive experiments on Hellomeme and HunyuanPortrait demonstrate that `StrucBooth` substantially improves expression similarity and structural consistency without compromising identity or perceptual quality. These results highlight the effectiveness of incorporating structural gradients as a supervisory signal, showing that a lightweight, few-step optimization can significantly enhance expression fidelity in portrait generation.

ETHICS STATEMENT

Our work focuses on advancing portrait animation and expression generation to improve applications in education, communication, and creative media. We emphasize that our research is intended for constructive purposes and is not designed to deceive or mislead. Like all generative technologies, our methods could potentially be misused; we firmly oppose any use that could create harmful or deceptive content, such as impersonating real individuals without consent.

All data used in our work are either publicly available or synthetically generated. Specifically, the human faces and expressions are sourced from publicly accessible meme collections, fun caricatures, or AIGC-generated portraits. No private, proprietary, or non-consensual personal data were used.

REPRODUCIBILITY STATEMENT

We have made every effort to ensure the reproducibility of our results. Detailed descriptions of the training objectives, model architectures, and optimization settings are provided in the main paper and Appendix. Furthermore, we commit to releasing our code and trained models upon acceptance to promote transparency and reproducibility.

LLM USAGE STATEMENT

In this work, Large Language Models (LLMs) were used solely as a writing assistant to polish the manuscript, such as improving grammar, clarity, and style. They were not used for research ideation, experimental design, implementation, data analysis, or result interpretation. All technical contributions, experiments, and conclusions are entirely the responsibility of the authors.

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

# 6 APPENDIX

## 6.1 BASELINES

**HelloMeme.** The HelloMeme baseline is built upon a pre-trained Stable Diffusion (SD1.5) model and incorporates additional modules to enable fine-grained control over portrait generation. Its main principle is to extract fidelity-rich conditions, including facial expressions and head pose, and inject them into the generative pipeline to guide the synthesis of target images or videos. The model operates in a two-stage manner, where the first stage captures coarse facial and pose information, and the second stage refines the output to match the driving conditions. This design allows the model to preserve identity and expression details while leveraging the generative power of the underlying diffusion backbone. To protect identity information during training, strong perturbations such as random blurring are applied to sensitive regions like the eyes and mouth, ensuring the network learns generalizable mappings rather than memorizing specific facial features.

**Hunyuan Portrait.** The Hunyuan Portrait baseline addresses the challenges of diverse facial geometries and intricate expression details by proposing an implicit conditional control framework. It uses stable video diffusion as the backbone and integrates identity and motion information through appearance and motion attention, avoiding the need for fine-tuning the image diffusion model or separately training a motion module. Identity and motion are coarsely decoupled using a pre-trained motion encoder, followed by enhanced training strategies and improved network architectures to strengthen motion control and portrait identity separation. To better model temporal dependencies in video generation, a motion memory bank is incorporated to provide an implicit representation of motion features. An intensity-aware motion encoder is introduced to handle variations in motion blur and pixel distortions, capturing fine-grained motion details. Additionally, consistent modeling of portrait identity and background is achieved by combining ArcFace with a DiNOv2 backbone to build an enhanced appearance encoder.

## 6.2 IMPLEMENTATION DETAILS

**Method Details.** For the face detector,we use face-alignment library to detect 2D facial landmarks, extract the corresponding areas related to left eye, right eye, mouth, eyebrows, etc... Bounding boxes covering the current expression regions are obtained according to the positions of the corresponding landmarks and are expanded by roughly 5–25 pixels to ensure that key structural information is included without introducing unnecessary content.

We first detect the PGT image to obtain a rough estimate of the expression regions in the predicted result. These bounding boxes are then scaled down to match the latent space, allowing us to crop the corresponding patches from the latent representation and decode them. For the driving image, we similarly detect the facial expression regions and resize the cropped regions according to the patch size from the decoded prediction, ensuring that their dimensions are consistent.

We then transfer all these patches into grayscale:

$$I_{\text{gray}} = 0.2989 \cdot R + 0.5870 \cdot G + 0.1140 \cdot B, \tag{9}$$

then we extract image gradients using Sobel convolution kernels. For a given input image $I$, the horizontal and vertical gradients are computed as:

$$G_x = K_x * I, \quad G_y = K_y * I, \tag{10}$$

where the Sobel kernels $K_x$ and $K_y$ are defined as:

$$K_x = \begin{bmatrix} 1 & 0 & -1 \\ 2 & 0 & -2 \\ 1 & 0 & -1 \end{bmatrix}, \quad K_y = \begin{bmatrix} 1 & 2 & 1 \\ 0 & 0 & 0 \\ -1 & -2 & -1 \end{bmatrix}. \tag{11}$$

**Experiment Details.** All baseline fine-tuning experiments were conducted using an 8-rank LoRA setup with a learning rate of 2e-5, employing the AdamW optimizer with a constant schedule. Both images and videos were processed at a resolution of $512 \times 512$. For HelloMeme, we adopted the v3 version and used the RV module as the base generator. For Hunyuan Portrait, we used the recommended model weights provided by the authors. Specifically, during training, the LoRA modules

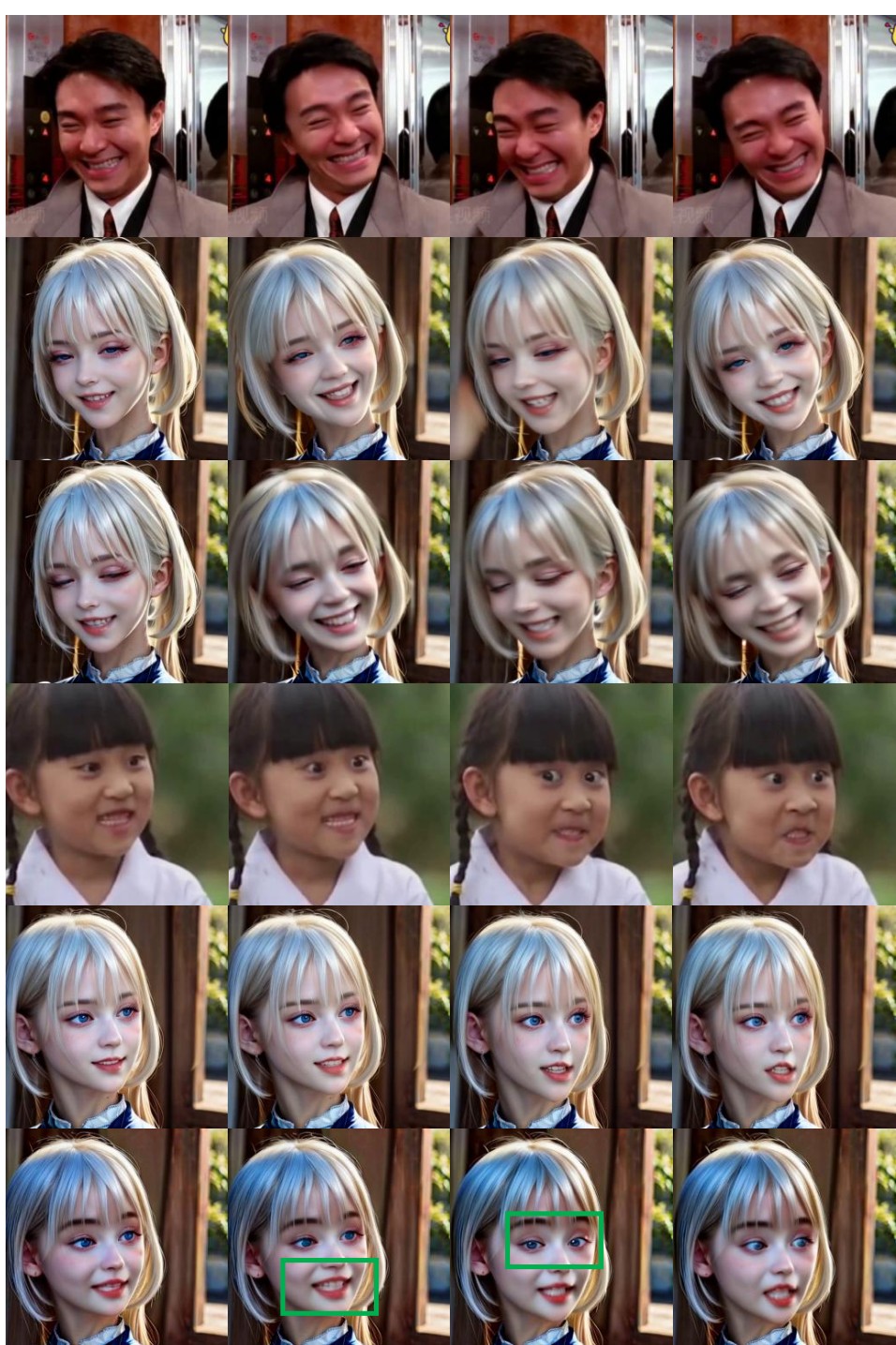

Figure 6: **More results of Videos on HelloMeme.** We show more results of HelloMeme. For each three lines, from top to bottom is driven frames, baseline frames and our frames.

were updated across a full timestep range of 0–1000. During inference, for the 25-step generation process, LoRA was disabled in the last 15 steps to avoid overfitting or interference from fine-tuned weights in the final refinement stage.

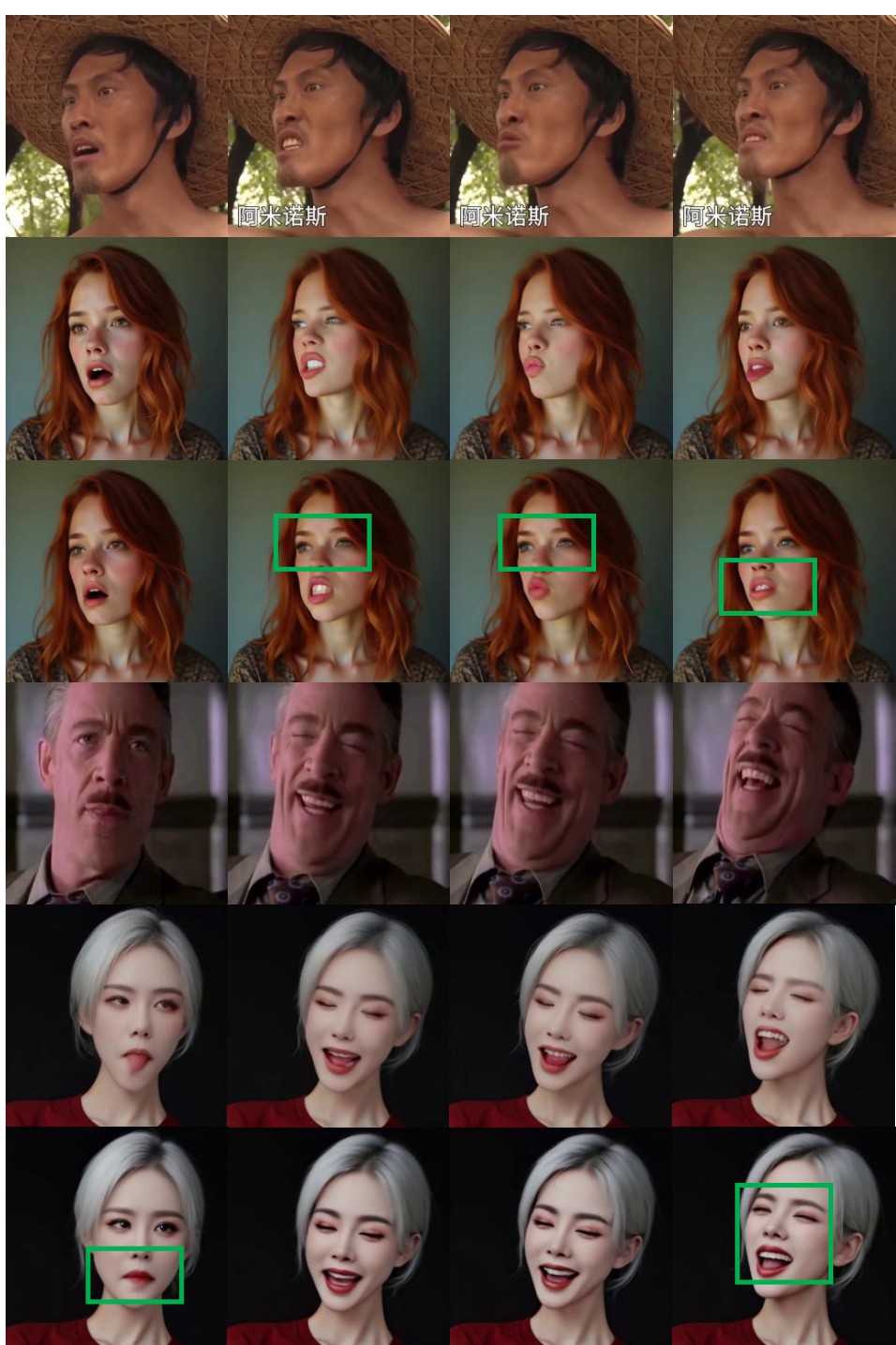

Figure 7: **More results of HunyuanPortrait.** We show more results of HunyuanPortrait. For each three lines, from top to bottom is driven frames, baseline frames and our frames.

## 6.3 MORE RESULTS.

**HelloMeme on Cross-Video.** We present additional results of HelloMeme on the Cross-Video Bench in Fig. 6. In the first three cases, the baseline synthesis produces results with eyes remaining open and insufficient mouth opening, whereas our method achieves better fidelity in details related

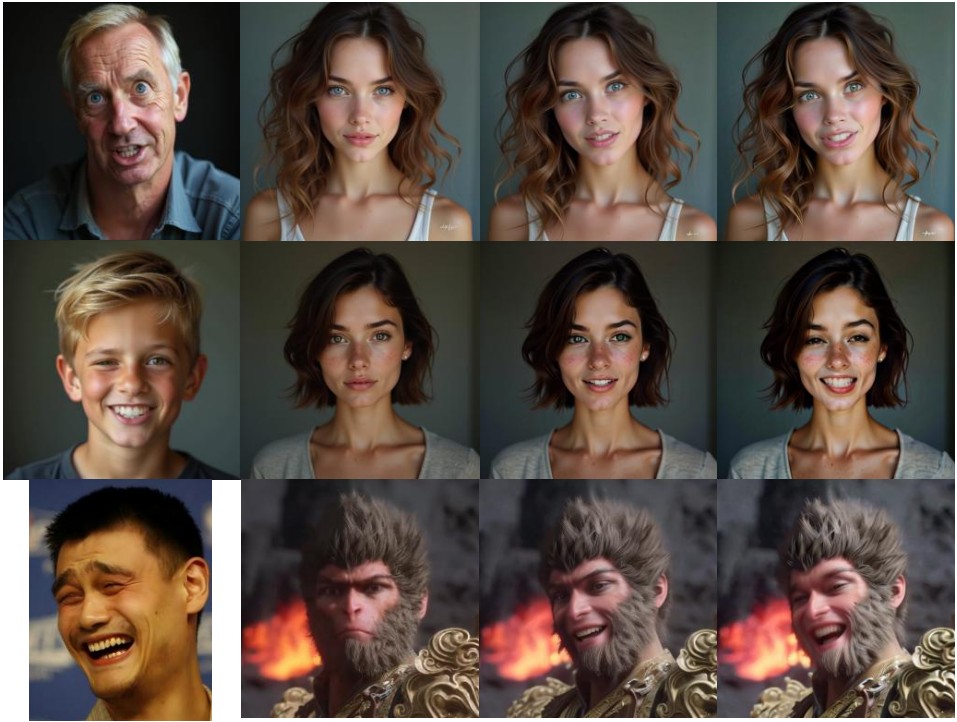

Figure 8: **More results of images on HelloMeme.** We show more results of HelloMeme on Cross-Image Bench. For each line, from left to right represents drive image, identity image, hellomeme result and our result.

to eye closure and smiling. In the latter three cases, our approach generates mouth shapes and expressions with greater strength and closer resemblance to the driving images. Moreover, in the third frame shown, the baseline fails to reasonably shift the eye gaze, while our optimized method addresses this issue. These comparisons demonstrate that, relative to the weaker HelloMeme baseline, our method can effectively enhance the similarity of expression synthesis.

**HunyuanPortrait on Cross-Video.** We also present the results of HunyuanPortrait on the Cross-Video Bench in Fig. 7. Improvements over the baseline are highlighted with green boxes. As shown, our optimizations enhance the synthesis capability of the base model, particularly in details such as eye gaze and mouth shape.

**HelloMeme on Cross-Image** We also present results on the Cross-Image Bench in Fig. 8. In each row, from left to right, we show the driving image, the identity reference image, the synthesis result of HelloMeme, and the synthesis result of our method. It can be observed that our synthesized expressions are closer to the driving image in terms of structural details and expression amplitude.

## 6.4 LIMITATION

Although our method enhances the accuracy of expression imitation by completing structural information, the quality of the simulated expressions and overall image fidelity remain constrained by the performance of the underlying generator. In addition, for overly blurred images or cases where the face cannot be recognized, our method struggles to extract valid regions or aligned structures, which limits its effectiveness in extreme scenarios. Furthermore, in expression composition applications, if the poses of different reference expressions differ significantly, the transfer process may suffer from severe misalignment, making it difficult to obtain satisfactory results.

