# OpenReview forum: "StrucBooth: Structural Gradient Supervised Tuning for Enhanced Portrait Animation"
_ICLR.cc/2026/Conference — ICLR 2026 Conference Withdrawn Submission_

### Official Review · Reviewer_sRb9 · 2025-10-25

**Soundness:** 2
**Presentation:** 3
**Contribution:** 2
**Rating:** 2
**Confidence:** 4

**Summary:**

This paper proposes StrucBooth, a framework designed to address a critical limitation in existing portrait animation methods: the loss of fine-grained facial structural details (e.g., eye contour, eyebrow shape, mouth deformation) when transferring expressions from a reference (driving) image/video to a target identity. Existing approaches rely on high-level expression encoders or 2D/3D feature representations, which capture only coarse semantics and omit pixel-space structural cues—leading to noticeable expression discrepancies. StrucBooth resolves this via two core components: PGT-based Self-Tuning and Facial Structural Gradients Supervision. The authors validate StrucBooth on three benchmarks (Cross-Image, Cross-Video, Self-Video) using two baselines (HelloMeme, HunyuanPortrait). Quantitative results show consistent improvements.

**Strengths:**

- Unlike methods that rely on latent-space structural control (ControlNet, Zhang et al. 2023a) or 3D morphable models (3DMM, Doukas et al. 2021), StrucBooth directly leverages pixel-level gradients. This preserves critical local structures (e.g., eye gaze direction, mouth curvature) that are lost in coarse feature encodings—addressing a longstanding gap in the field.
- The PGT-based self-tuning avoids the need for large-scale retraining or external supervision, a limitation of methods like EchoMimic (Chen et al. 2024b) which require landmark preprocessing. Additionally, patch-wise gradient computation reduces GPU memory usage and speeds up training by ~22% (per ablation), enabling video-level optimization (unlike global gradient methods, e.g., X-Portrait, Xie et al. 2024).
- The proposed PasteDrive module allows fusing expression details from multiple references (e.g., combining a smile from one reference and raised eyebrows from another). This extends beyond the single-reference transfer of SOTA works (e.g., SadTalker, Zhang et al. 2023b) and unlocks creative applications.
- Evaluations across static (Cross-Image) and dynamic (Cross-Video, Self-Video) scenarios, with both objective (EXP SIM, AED) and qualitative metrics, ensure generalizability. Most competing works (e.g., ControlNext, Peng et al. 2024) focus solely on video or image animation, limiting their scope.

**Weaknesses:**

- While the combination of PGT and gradient supervision is practical, individual components lack groundbreaking innovation. Sobel-based gradient extraction is a standard computer vision technique, and PGT resembles self-supervised fine-tuning (e.g., used in latent video diffusion, He et al. 2022). The paper provides no theoretical analysis of why this specific combination outperforms alternatives (e.g., Canny edges, texture maps) or why PGT is superior to other self-supervision signals (e.g., contrastive learning).
- The authors only compare against two baselines (HelloMeme, HunyuanPortrait), excluding key SOTA methods like SadTalker (audio-driven animation), EchoMimic (landmark-guided refinement), or Memo (memory-guided video diffusion, Zheng et al. 2024). This makes it impossible to contextualize StrucBooth’s performance—for example, how it fares against methods optimized for extreme expressions or low-quality inputs.
- The ablation study explores optimization steps and patch-wise decoding but ignores critical hyperparameters (e.g., LoRA rank, learning rate, loss weights λₗₙₘ, λ₉ᵣₐ𝒹, λᵢ𝒹) and their impact on performance. Additionally, the trade-off between expression similarity and identity preservation is mentioned but not quantified (e.g., no Pareto frontier analysis across loss weight combinations).
- All experiments rely on objective metrics, but portrait animation’s utility depends on human perception of naturalness. SOTA works like EmoNet include user studies to validate subjective quality—an absence here that weakens claims of “visually consistent” results.
- StrucBooth cannot overcome flaws in the underlying generator (e.g., HelloMeme’s SD1.5 backbone). If the base model fails to generate plausible facial structures initially, StrucBooth cannot recover fine details—unlike methods like Megaportraits (Drobyshev et al. 2023), which integrate super-resolution to enhance input quality.
- For blurred images or unrecognizable faces (failed face detection/landmark extraction), StrucBooth cannot extract valid expression patches. Competing works (e.g., HunyuanPortrait, Xu et al. 2025) use robust appearance encoders to mitigate this, but StrucBooth lacks such preprocessing.
- When combining expressions from references with large pose differences, StrucBooth suffers from misalignment. Methods like MimicMotion (Zhang et al. 2024b) use pose-aware guidance to avoid this, but StrucBooth does not address pose consistency in composition.
- While evaluated on 3-second clips, StrucBooth lacks explicit temporal modeling (e.g., factorized spatiotemporal attention in VideoCrafter2, Chen et al. 2024a). This may lead to flickering in longer videos—a gap not discussed in the paper.

**Questions:**

- Why was the Sobel operator chosen for gradient extraction over alternatives (e.g., Prewitt, Canny, or learnable gradient modules)? Were experiments conducted to validate that Sobel gradients best capture expression-related structures?
- How would StrucBooth perform on datasets with more diverse identities (e.g., VoxCeleb2, CelebA-HQ: varying ages, ethnicities) or extreme expressions (e.g., crying, yawning)? The current test set (10 identities, 20 driving inputs) is too small to confirm generalizability.
- Could integrating preprocessing modules (e.g., face super-resolution, robust landmark detection) address low-quality input issues? Would adding 3DMM-based pose alignment resolve misalignment in compositional control?
- The “sparse identity loss” is mentioned but not detailed. How does it compare to SOTA identity preservation techniques (e.g., ArcFace in HunyuanPortrait) in terms of effectiveness and computational cost?

---

### Official Review · Reviewer_8y3J · 2025-10-28

**Soundness:** 3
**Presentation:** 2
**Contribution:** 2
**Rating:** 2
**Confidence:** 5

**Summary:**

The paper introduces StrucBooth, a two-stage pipeline for transferring head pose and facial expressions from a reference image to a target while preserving identity. Stage 1 performs an initial pose/expression adaptation via LoRA-based fine-tuning. Stage 2 further refines expression by optimizing the target image with gradients computed from the reference.

**Strengths:**

The idea of gradient-guided expression refinement is interesting and potentially useful for facial reenactment.

The two-stage design is simple to implement and could be compatible with existing diffusion backbones.

**Weaknesses:**

Major Concerns.

- The PGT/LoRA fine-tuning with added noise/denoising appears incremental and closely mirrors well-known practices. As presented, this stage reads as engineering rather than a substantive algorithmic contribution; clearer positioning versus standard LoRA fine-tuning baselines is needed.
- While gradient guidance is appealing, the reliance on manually defined, patch-based regularization raises key issues: (i) expression cues are not confined to eyes/mouth—many are diffuse or coupled with global facial changes—so hand-selecting regions risks omission or leakage; (ii) larger masks exacerbate shape-mismatch artifacts between reference and target, which seem visible as distortions/blur in Figs. 3–4. A more principled region selection or deformation model appears necessary.

Minor Concerns.
- How are ``expression-related'' latent patches identified? Please detail the procedure and provide ablations.
- Simple resizing is unlikely to handle identity/shape differences. Consider facial landmarks, dense flow/optical flow, 3DMM-based warping, or TPS to reduce artifacts.
- The Sobel kernel description (Eqs. 3–4) is not essential; space could be used for core technical insights or ablations.
- Compare against more SOTAs in face reenactment/pose-expression transfer.

**Questions:**

Please see the weakness

---

### Official Review · Reviewer_UFXF · 2025-11-01

**Soundness:** 2
**Presentation:** 3
**Contribution:** 2
**Rating:** 2
**Confidence:** 4

**Summary:**

The paper presents StrucBooth, a case-specific fine-tuning framework designed to enhance portrait animation by integrating pixel-level structural gradient supervision. The central idea is that existing cross-identity portrait animation models often lose subtle expression nuances—such as eye closure or mouth curvature—because they rely primarily on high-level semantic representations (e.g., latent expression codes). StrucBooth addresses this limitation by embedding fine-grained facial structure cues directly into the model through a few-step self-tuning process guided by facial structural gradients.

**Strengths:**

StrucBooth introduces a novel, case-specific fine-tuning framework that integrates pixel-level structural gradient supervision to enhance expression fidelity in portrait animation while preserving identity. Its patch-wise supervision allows fine-grained, controllable expression refinement, and extensive experiments with detailed ablations convincingly demonstrate consistent gains in both image- and video-based settings.

**Weaknesses:**

1. The method relies heavily on accurate face detection and patch-wise gradient extraction, yet the paper does not discuss what happens when detection fails or when the driving frame is occluded or misaligned, raising concerns about robustness in unconstrained conditions
2. If the PGT is poorly generated, the optimization could reinforce incorrect facial structures, but this dependency is not thoroughly analyzed.
3. Identity leakage from the driving expression remains a noticeable issue, as the paper does not quantify how structural alignment influences identity preservation; from the visual examples (e.g., the upper-left case in Fig. 1), leakage appears in multiple instances.
4. is the L_{grad} applied to all timesteps t during optimization? If large t values are included, the predicted x_0 would be highly noisy, potentially leading to unstable supervision.
5. The motivation for pixel-level structural supervision also needs deeper justification. Although the paper argues that prior works relying on high-level expression encoders overlook fine-grained details, different identities should naturally express the same emotion with stylistic variation—strict pixel-level matching might constrain this diversity and hinder natural adaptation.
6. The definition of a “case” is ambiguous—whether it refers to a single image, a video sequence, or all clips of one identity—and the presence of multiple balancing coefficients further complicates scalability to large-scale application.
7. The improvement over the base model is sometimes marginal; for example, in the third row of Fig. 5, the expression accuracy around the mouth region remains unsatisfactory.
8. no video results are provided, making it difficult to assess temporal consistency or animation quality. Additional experiments with extreme expressions and pose variations would significantly strengthen the paper’s claims.

**Questions:**

please check the weakness part.

---

### Official Review · Reviewer_jk16 · 2025-11-04

**Soundness:** 2
**Presentation:** 3
**Contribution:** 1
**Rating:** 2
**Confidence:** 4

**Summary:**

This paper proposes a method for face reanimation built on top of pretrained diffusion models using LoRA. The central contributions of this paper are the following

1. Use of a Psuedo GroundTruth generated by the model as supervision. More specifically, the non-LoRA version of the model (ie the base model) generates an estimate that is regressed to via LoRA.
2. The LoRA is also trained using local gradients from the expression relevant regions of the driving image to enforce expression consistency.

Quantitative and Qualitative results do show minor improvements over the baselines in terms of expression adherence.

**Strengths:**

1. The methodology of the paper is well motivated
2. The paper is easy to read and follow
3. The experiments and ablations are relatively complete

**Weaknesses:**

1. The qualitative results show a massive identity shift in the Cross image case, which is also seeming reflected in the quantitative results (Table 1). This yields results that are worse than the baseline in my opinion. Unfortunately this is major weakness of this method

**Questions:**

None

---

### Note · Authors · 2025-11-12

I have read and agree with the venue's withdrawal policy on behalf of myself and my co-authors.